# Motion Capture Technology in Industrial Applications: A Systematic Review

**DOI:** 10.3390/s20195687

**Published:** 2020-10-05

**Authors:** Matteo Menolotto, Dimitrios-Sokratis Komaris, Salvatore Tedesco, Brendan O’Flynn, Michael Walsh

**Affiliations:** Tyndall National Institute, University College Cork, T23 Cork, Ireland; salvatore.tedesco@tyndall.ie (S.T.); brendan.oflynn@tyndall.ie (B.O.); michael.walsh@tyndall.ie (M.W.)

**Keywords:** health and safety, IMU, industry 4.0, motion tracking, robot control, wearable sensors

## Abstract

The rapid technological advancements of Industry 4.0 have opened up new vectors for novel industrial processes that require advanced sensing solutions for their realization. Motion capture (MoCap) sensors, such as visual cameras and inertial measurement units (IMUs), are frequently adopted in industrial settings to support solutions in robotics, additive manufacturing, teleworking and human safety. This review synthesizes and evaluates studies investigating the use of MoCap technologies in industry-related research. A search was performed in the Embase, Scopus, Web of Science and Google Scholar. Only studies in English, from 2015 onwards, on primary and secondary industrial applications were considered. The quality of the articles was appraised with the AXIS tool. Studies were categorized based on type of used sensors, beneficiary industry sector, and type of application. Study characteristics, key methods and findings were also summarized. In total, 1682 records were identified, and 59 were included in this review. Twenty-one and 38 studies were assessed as being prone to medium and low risks of bias, respectively. Camera-based sensors and IMUs were used in 40% and 70% of the studies, respectively. Construction (30.5%), robotics (15.3%) and automotive (10.2%) were the most researched industry sectors, whilst health and safety (64.4%) and the improvement of industrial processes or products (17%) were the most targeted applications. Inertial sensors were the first choice for industrial MoCap applications. Camera-based MoCap systems performed better in robotic applications, but camera obstructions caused by workers and machinery was the most challenging issue. Advancements in machine learning algorithms have been shown to increase the capabilities of MoCap systems in applications such as activity and fatigue detection as well as tool condition monitoring and object recognition.

## 1. Introduction

Motion capture (MoCap) is the process of digitally tracking and recoding the movements of objects or living beings in space. Different technologies and techniques have been developed to capture motion. Camera-based systems with infrared (IR) cameras, for example, can be used to triangulate the location of retroreflective rigid bodies attached to the targeted subject. Depth sensitive cameras, projecting light towards an object, can estimate depth based on the time delay from light emission to backscattered light detection [1]. Systems based on inertial sensors [2], electromagnetic fields [3] and potentiometers that track the relative movements of articulated structures [4] also exist. Hybrid systems combine different MoCap technologies in order to improve precision and reduce camera occlusions [5]. Research has also focused on the handling and processing of high dimensional data sets with a wide range of analysis techniques, such as machine learning [6], Kalman filters [7], hierarchical clustering [8] and more.

Thanks to their versatility, MoCap technologies are employed in a wide range of applications. In healthcare and clinical settings, they aid in the diagnosis and treatment of physical ailments, for example, by reviewing the motor function of a patient or by comparing past recordings to see if a rehabilitation approach had the desired effect [9]. Sports applications also benefit from MoCap by breaking down the athletes’ motion to analyse the efficiency of the athletic posture and make performance-enhancing modifications [10]. In industrial settings, MoCap is predominately used in the entertainment [11] and gaming industry [12], followed by relatively few industrial applications in the sectors of robotics [13], automotive [14] and construction [15]. However, the need for highly specialised equipment, regular calibration routines, limited capture volumes, inconvenient markers or specialised suits, as well as the significant installation and operation costs of MoCap systems, has greatly impeded the adoption of such technologies in other primary (i.e., extraction of raw materials and energy production) and secondary industrial applications (i.e., manufacturing and construction). Nevertheless, the fourth industrial revolution has brought new forms of industrial processes that require advanced and smart sensing solutions; as MoCap technology becomes more convenient and affordable [16], and applicable in challenging environments [17], its application becomes more attractive for a wider range of industrial scenarios.

Since industrial technologies are constantly changing and evolving in order to meet the demands of different sectors, it is important to track the technological progress and the new trends in hardware and software advancements. Previous reviews have focused on MoCap in robotics [13], clinical therapy and rehabilitation [18], computer animation [12], and sports [19]; however, the use of MoCap for industrial applications has not been yet recorded in a systematic way. The purpose of this work is to report on the development and application of different commercial and bespoke MoCap solutions in industrial settings, present the sectors that mainly benefit from them (e.g., robotics and construction), and identify the most targeted applications (e.g., infrastructure monitoring and workers’ health and safety). Along these lines, this review aims to provide insight on the capabilities (e.g., robust pose estimation) and limitations (e.g., noise and obstructions) of MoCap solution in industry, along with the data analytics and machine learning solutions that are used in conjunction with MoCap technologies in order to improve the potency of the sensors, support in the processing of large quantities of output data and aid in decision-making processes.

## 2. Materials and Methods

### 2.1. Search Strategy

This study was aligned with the Preferred Reported Item for Systematic review and Meta-Analysis (PRISMA) statement [20]. A literature search was carried out on Embase, Scopus and Web of Science databases from the 9th to the 16th of March 2020. Titles, abstracts and authors’ keywords were screened with a four-component search string using Boolean operators. The first three components of the string were linked with AND operators and were formed of keywords and their spelling variations that are associated with motion analysis (e.g., biomechanics, kinematics, position), the sensors used to capture motion (e.g., IMUs), and the industrial setting (e.g., industry, occupation, factory), respectively. A NOT operator preceded the fourth section of the string that was a concatenation of terms detached from the aims of this review (e.g., surgery, therapy, sports, animals). Google Scholar was also employed to screen for keywords in abstracts that were published up to one year prior to the literature search. The full search strings used in Google Scholar and for each database search are also included in Appendix A.

Studies met the inclusion criteria if they were written in English and were published from January 2015 onwards. The search included both in press and issued articles that were published in scientific journals or conference proceedings alike. Review papers and conference abstracts were all excluded from this work since they do not typically report on all elements of the predefined tables that were used for data extraction. To ensure consistency of comparison, only studies that actively employed sensors that are designed to directly measure motion (i.e., the position, displacement, velocity or acceleration of an object) for either primary and secondary industrial applications were included; in this context, an industrial application was defined as any process related to the extraction of raw materials (e.g., metals or farming), or the manufacturing and assembly of goods (e.g., cars or buildings). Therefore, proof of concept papers that were not tested experimentally, simulations, and studies concerning white collar workers (e.g., office or other non-manual workers) and were excluded; additionally, works employing sensors that can indirectly measure motion (e.g., electromyography (EMG) in conjunction with machine learning algorithms [21]) were also omitted. Articles were included only if the participants’ sample size (where applicable), and the type, number and placement of all used sensors were reported. Journal papers were prioritized in the event where their contents were also covered in earlier conference publications; in cases where this overlap was only partial, multiple publications were included.

All articles were imported to a standard software for publishing and managing bibliographies and duplicates were automatically removed. Two independent reviewers screened all titles and abstracts and labelled each article based on its conformity with the aims of the study. Articles that both reviewers deemed as non-compliant with the predefined inclusion criteria were excluded from further review. The remaining articles were then fully screened, and each reviewer provided reasons for every exclusion. Conflicts between the reviewers were debated until both parties agreed to a conclusion. Finally, the reference lists of all approved articles were browsed to discover eligible articles that were not previously found; once more, both reviewers individually performed full-text screenings and evaluated all newly found publications.

### 2.2. Assessment of Risk of Bias

Two reviewers assessed the risk of bias and the quality of all considered studies using an adapted version of the AXIS appraisal tool for cross-sectional studies [22]. Questions 6, 7, 13, 14, 15 and 20 of the original AXIS appraisal tool were disregarded since they assess issues that are not often apparent in studies concerning industrial applications, such as taking samples from representative populations, non-responders, non-response bias and participants’ consent. The remaining elements of the AXIS list were adapted to form twelve questions that could be answered with a “yes” or a “no” and were used to appraise each study (Table 1) by summing all affirmative responses and providing a concluding score out of 12. Studies ranked below 6 were viewed as having a high risk of bias, while studies with ratings over 7 and 10 were considered of medium or low risk, respectively. The average study ratings of both reviewers were also computed to confirm the inter-rater assessment consistency.

### 2.3. Data Extraction

Data from all considered articles were extracted by each reviewer independently using predefined tables. Cumulative logged data were systematically monitored by both parties to ensure coherent data collection. Authors’ names, year of publication, sample size, sensor placement (binned: machinery, upper, lower or full body), number and types of all used sensors (e.g., IMU or cameras), secondary validation systems (where applicable), and main findings were all recorded. In reference to their respective broad objectives, all considered articles were allocated into four groups based on whether they aimed to ensure workers’ health and safety (e.g., postural assessment, preventing musculoskeletal injuries, detecting trips and falls), to directly increase workers’ productivity (e.g., workers’ location and walk path analysis), to conduct machinery monitoring and quality control (e.g., cutting tool inspections), or to improve an industrial process (e.g., hybrid assembly systems) or the design of a product (e.g., car seats). If a work could fall into more than one category [23] (e.g., health and safety, and workers’ productivity), the paper was allocated in the most prominent category. Additionally, the directly beneficiary industry sector was recorded (e.g., construction, aerospace, automotive, or energy); in the instance of a widespread application, the corresponding article was labelled as “generic”. Studies that employed machine learning were additionally logged, along with the used algorithm, type of input data, training dataset, output and performance.

## 3. Results

### 3.1. Search Results

Database searching returned 1682 records (Figure 1). After removing duplicates (n = 185), the titles, keywords and abstracts of 1497 articles were screened and 1353 records were excluded as they did not meet inclusion criteria. The remaining articles (n = 144) were assessed for eligibility, and 47 papers were retained in the final analysis. Twelve more records were added after screening the reference lists of the eligible papers, bringing the total number of the included studies to 59. Four, 13 and 16 records were published in 2015, annually from 2016 to 2018, and in 2019, respectively, underlying the increasing interest of the research community on the topic.

### 3.2. Risk Assessment

Twenty-one and 38 studies were assessed as being prone to medium and low risks of bias, respectively (Table 2). None of the considered articles scored lower that six on the employed appraisal checklist. All reviewed articles presented reliable measurements (Q5) and conclusions that were justified by their results (Q10); yet, many authors have inadequately reported or justified sample characteristics (Q3, 37%), study limitations (Q11, 53%) and funding or possible conflict sources (Q12, 51%). Statistics (Q6, 81%) and general methods (Q7, 88%) were typically described in depth. Generally, studies were favourably assessed against all the remaining items of the employed appraisal tool (Q1, 95%; Q2, 92%; Q4, 93%; Q8, 98%; Q9, 93%). The assessments of both reviewers were consistent and comparable with average review scores of 9.9 ± 1.6 and 9.9 ± 0.9.

### 3.3. MoCap Technologies in Industry

In the reviewed studies, pose and position estimation was carried out with either inertial or camera-based sensors (i.e., RGB, infrared, depth or optical cameras), or in combination with each other (Table 3). Inertial sensors have been widely employed across all industry sectors (49.2% of the reviewed works), whether the tracked object was an automated tool, the end effector of a robot [30,37,64], or the operator [27,36,39]. In 30.5% of the reviewed studies, camera-based off-the-shelf devices such as RGB, IR and depth cameras, mostly coming from the gaming industry (e.g., Microsoft Kinect and Xbox 360), were successfully employed for human activity tracking, and gesture or posture classification [25,77]. Inertial and camera-based sensors were used in synergy in 10.2% of the considered works, in the tracking of the operator’s body during labour or the operator’s interaction with an automated system (e.g., robotic arm). EMG, ultra-wide band (UWB) nets, resistive bending sensors or scanning sonars were used along with IMUs to improve pose and position estimation in five studies (8.5%). One study also coupled an IMU sensor with a CCTV and radio measurements. Generally, IMU and camera-based sensors were used consistently in the industry during the last 5 years (Figure 2).

Considering that the most frequently adopted sensors used in industry were IMUs (e.g., Xsens MVN) and marker-based or marker-less (e.g., Kinect) camera systems, their characteristics, advantages and disadvantages were also mapped (Table 4) in order to evaluate how each sensors type is appropriate to the different applications. Naturally, the characteristics of each system vary greatly depending on the number, placement, settings and calibration requirements of the sensors, yet, general recommendations can be made for the adoption of a particular type of sensor for distinct tasks. Additionally, given the required level of accuracy, capture volume, budget and workplace limitations or other considerations, Table 4 shows the specifications and most favoured industrial applications for each type of sensor (e.g., activity recognition, or human–robot collaboration).

### 3.4. Types of Industry Sectors

Most frequently, MoCap technologies were adopted by the construction industry (Table 5, 30.5%), followed by applications on the improvement of industrial robots (22%), automotive and bicycle manufacturing (10.2%), and agriculture and timber (8.5%). On a few occasions, authors engaged in applications in the food (5.1%) and aerospace industries (3.4%), while energy, petroleum and steel industries were each discussed in a single study (1.7%). All remaining applications were considered as generic (22%) with typical examples of studies monitoring physical fatigue [48,71], posture [45] and neck-shoulder pain [74] in workers. Construction, generic and robotic applications were the only researched topics in 2015, while automotive, agriculture and food industrial applications were explored every year after 2016; MoCap technologies in the aerospace, energy, steel and petroleum industries were disseminated only recently (Figure 3, left).

### 3.5. MoCap Industrial Applications

MoCap techniques for industrial applications were primarily used for the assessment of health and safety risks in the working environment (Table 6, 64.4%), whilst fatigue and proper posture were the most targeted issues [48,49,72]. The research interest of the industry in health and safety MoCap applications increased steadily over the reviewed period (Figure 3, right). Productivity evaluation was the second most widespread application (20.3%), with studies typically aiming to identify inefficiency or alternative approaches to improve industrial processes. Similarly, MoCap techniques were also employed to directly improve workers productivity (10.1%), whereas 8.5 % of the studies focused on task monitoring [17] or in the quality control of an industrial processes [30].

### 3.6. MoCap Data Processing

In the majority of the reviewed works, raw sensor recordings were subject to data synchronization, pre-processing, and classification. Data synchronisation was occasionally reported as part of the pre-processing stage and included in the data fusion algorithm [24,34,36], but technical details were frequently omitted in the reviewed studies [27,28]; yet, when the synchronization strategy was reported, a master control unit [36,50,51,54] or a common communication network [15,31,67] were used. Different sampling rates of data streams were addressed by linear interpolation and cross-correlation [73] techniques, or by introducing a known event that triggers all the sensors [29,47,49,55].

In the pre-processing stage, data were filtered to mitigate noise and address drift, outliers and missing points in data streams (to avoid de-synchronisation for instance), then were fused together, and were further processed to extract the numerical values of interest; in the studies considered by this review, this was mostly achieved via low-pass filters (e.g., nth order Butterworth, sliding window and median filters) [15,31,40,45,58,61,62,63,66,69,73,75,77], Kalman filters [17,28,29,40,41,51,54,57,60,71,73] and band-pass filters when EMG data were collected [29,49,50,51,54]. The drift of inertial data, a typical inertial sensors issue, was sometimes addressed in the pre-processing stage by implementing filtering methods such as the zero-velocity update technique [44,59,60].

Data classification was obtained by establishing thresholds or via machine learning classifiers. An example of threshold was given by [39], where trunk flexion of over 90° was selected to identify a high ergonomic risk, or by [31] where the position of the operator’s centre of mass and the increasing palm pressure identified a reach-and-pick task. Such thresholds were obtained based on observations or established models and standards (e.g., RULA: Rapid Upper Limb Assessment, and REBA: Rapid Entire Body Assessment scores). Machine learning techniques were employed in 18.6% of the reviewed works (Table 7), aiming to build an unsupervised or semi-supervised system able to improve its own robustness and accuracy while increasing the number of outcomes that were correctly predicted. The most used algorithms were Artificial Neural Network (ANN), Support Vector Machine (SVM) and Random Forest (RF), with ANN and SVM being mostly employed for binary or three group classification, while random forest for multiclass classification. The accuracy of the developed machine learning algorithms typically ranged from 93% to 99% (Table 7).

### 3.7. Study Designs and Accuracy Assessments

Overall, the reviewed studies dealt with small sample sizes of less than twenty participants, with the exception of Tao et al. [56], Muller et al. [38] and Hallman et al. [74] who recruited 29, 42 and 625 participants, respectively. Eighteen, 13 and 8 studies placed IMU sensors on the upper, full and lower body, respectively, while six authors attached IMUs on machinery (Table 8). Out of the 41 studies that employed inertial units (70% of all the works), the majority of the authors used less than three sensors (25 studies, Table 8), while seven groups used 17 sensors, as a part of a pre-developed biomechanical model with systems such as the Xsens MVN, to capture full body movements. Sensor placement for all the studies that did not adopt pre-developed models is graphically depicted on Figure 4. Six studies accompanied motion tracking technologies with EMG sensors [29,49,50,51,54,57], two with force plates [73,75], two with pressure mats [61,62] and one with instrumented shoes [73]. Two works also used the Oculus Rift virtual reality headset to remotely assess industrials locations and control robotic elements [39,43]. The tracking accuracy of the developed systems was directly assessed against gold-standard MoCap systems (e.g., Vicon or Optotrack; Table 8, in bold) in six works [14,15,55,59,73,77], while the classification or identification accuracy of a process was frequently evaluated with visual inspection of video or phone cameras [15,29,36,44,60,63,69]. A thorough diagram showing the connections between type of industry, application and MoCap system, for each considered study is also presented on Figure 5.

## 4. Discussion

Industry 4.0 has introduced new processes that require advanced sensing solutions. The use of MoCap technologies in industry has been steadily increasing over the years, enabling the development of smart solutions that can provide advanced position estimation, aid in automated decision-making processes, improve infrastructure inspection, enable teleoperation, and increase the safety of human workers. The majority of the MoCap systems that were used in industry were IMU-based (in 70% of the studies, Table 3), whilst camera-based sensors were employed less frequently (40%), most likely due to their increased operational and processing cost, and other functional limitations, such as camera obstructions by workers and machinery which were reported as the most challenging issues [25,45,55]. Findings suggest that the selection of the optimal MoCap system to adopt was primarily driven by the type of application (Figure 5); for instance, monitoring and quality control was mainly achieved via IMUs sensors, while productivity improvement via camera-based (marker-less) systems. Type of industry was the second factor that had an impact on the choice of a MoCap system (Figure 5); for example, in a highly dynamic environment, like in construction sites, where the setup and configuration of the physical space change over time, wearable inertial sensors were the best option, since they could ensure a robust and continuous assessment of the operator’s health and safety. In industrial robot manufacturing instead, where the environmental constraints are known and constant, health and safety issues were primarily addressed by camera-based systems.

The increased use of IMUs promoted the development of advanced algorithmic methods (e.g., Kalman filters and machine learning, Table 7) for data processing and estimation of parameters that are not directly measured with inertial systems [17]. Optoelectronic technologies performed better and with higher tracking accuracy in human–robot collaboration tasks [33] and robot trajectory planning [32,46,52], due to the favourable conditions in such applications (e.g., the limited working volume and the known robot configurations) which allowed cameras to avoid obstructions. In general, hybrid systems that incorporate both vision and inertial sensors were found to have improved tracking performance in noisy and highly dynamic industrial environments, compensating for drift issues of inertial sensors and long-term occlusions which can effect camera-based systems [15,40]. For instance, in Papaioannou et al. [40], the trajectory tracking error caused by occlusion in a hybrid system was approximately half that of a camera-based tracking system.

Workers’ health and safety was found to be the most prolific research area. Even though wearable sensors are widely used in clinical settings for the remote monitoring of physiological parameters (e.g., heart rate, blood pressure, body temperature, VO2), only a single study [26] has employed multiple sensors for the measurement of such metrics in industrial scenarios. This can be attributed to the industries involved being interested in the prevention of work-related incidents that can lead to absence from work, rather than in the normative function of the workers’ body. As anticipated, health and safety research focused on the most common musculoskeletal conditions (e.g., back pain) and injuries (e.g., trips or injuries due to bad body posture), while the industries in which workers deal with heavy biomechanical loads or high risk of accidents (e.g., construction, Table 5) were the industries that drove the research. Fatigue and postural distress were also successfully detected by wearable inertial MoCap technologies [27,39,49,71,72]. When MoCap systems were combined with EMG sensors (Table 8), the musculoskeletal and postural evaluation of workers during generic physical activities (Table 5) was improved [29,48,49,50,51,54,57]. Inertial sensors also showed good results for the identification of hazardous events such as trips and falls in the construction industry [44,58,60,65,66,69,75], but the positions and numbers of the used IMUs were reported to impact on the intra-subject activity identification [26]. For example, fewer IMUs placed on specific anatomical sections (e.g., hip and neck) showed similar task classification performance than a greater number of IMUs distributed on the entire body [36]. In Kim et al. [36], a task classifier based on just two IMUs on the hip and head of the subject reached an accuracy of 0.7617 against the 0.7983 of the classifier based on 17 IMUs placed on the entire body. Activity recognition was also well performed by IMUs, and combined with activity duration measurements, made the evaluation of workers’ productivity in jobsites possible [24]. This topic was also the focus of interest for more than 10% of the studies in the past years (Table 6). However, when the assessment involved the identification or classification of tasks [26], secondary sensors were frequently needed in addition to the IMUs (force cells, temperature sensors, etc.).

Advancements were also reported in the development of efficient data classification algorithms that require large data streams, such as machine learning-based classifiers (Table 7). The usage of such algorithms has been documented in 11 works out of a total of 59, and was accompanied with a very high level of accuracy. The classification output of the reviewed algorithms differed greatly between the reviewed works, and covered applications from activity and fatigue detection to tool condition monitoring and object recognition (Table 7). However, the need of large training datasets, which usually require expert manual labelling to be produced, contradicted the very small sample sizes that were typically recruited (Table 8), and thus potentially impeding the broader use of machine learning beyond the proof-of-concept in applied cases in industry. The general lack of information regarding real-time capability of the presented classification algorithms was also identified as a potential drawback in real-world application, suggesting that more work is required to address this challenge. Yet, the reviewed works generally outlined the capacity of MoCap sensors in conjunction with machine learning solutions to provide solutions for activity recognition, tool inspection and ergonomic assessment in the workplace. These findings highlighted how the research activity on wearable systems for industrial applications is going towards solutions that can be easily embedded in working cloths. Improving key factors such as wearability, washability, battery duration, data storage and edge computing will be therefore essential. This improvement in the hardware design will have a direct impact on the amount and the quality of the data collection. This, as well, will have a beneficial effect on software development, especially for machine learning applications, were huge quantity of data are required. In this regard, attempts should be made for the further development and commercial distribution of processing algorithms that would improve the ease of use of such systems and the data processing.

Direct evaluation of the accuracy and tracking performance of a developed MoCap system [14,55] was generally achieved through comparisons with a high accuracy camera-based system. This is so far the most reliable process, as it guarantees an appropriate ground truth reference. However, the performance of algorithmic processes (e.g., evaluation of body postures or near-miss fall detections) was typically validated against visual observations of video recordings [69] or the ground truth that was provided by experts in the field [78], and therefore potentially biasing the accuracy of the respective method. As regards the use of commercially available MoCap solutions, a comparison was made of their limitations, advantages and applicability to industrial applications (Table 4) while the accuracy of off-the-shelf MoCap systems has been also extensively reviewed by van der Kruk and Reijne [82].

Even though all the reviewed works were assessed as being prone to medium and low risks of bias individually (Table 2), the main limitation at a study level was that more than half of the reviewed works (51%) did not properly report funding and conflict sources. This may be an indication of a critical source of bias, particularly in studies directly driven by the beneficiary industry, or in works that demonstrate MoCap systems that may be commercially available in the future. A limitation of this review stems from the potential publication bias and selective reporting across studies, which may affect the accumulation of thorough evidence in the field. Efforts from industry bodies to incorporate MoCap applications in their facilities that were either unsuccessful or were not disseminated in scientific journals were likely overlooked in this review. Finally, another limitation at a review-level arises from the short review period that narrowed the reporting of findings in a period of five years; however, the selected review period returned an adequate number of records for the justification of conclusions and exposure of trends (e.g., Figure 3), while also facilitating the reporting of multiple aspects of the reviewed articles, such as the studies’ design and key findings (Table 8).

## 5. Conclusions

This systematic review has highlighted how the industry 4.0 framework had led industrial environments to slowly incorporate MoCap solutions, mainly to improve the workers’ health and safety, increase productivity and improve an industrial process. Predominately, research was driven by the construction, robot manufacturing and automotive sectors. IMUs are still seen as the first choice for such applications, as they are relatively simple in their operation, cost effective, and present minimal impact on the industrial workflow in such scenarios. Moreover, inertial sensors have acquired, over the years, the performance (e.g., low power consumption, modularity) and size requirements to also be applied for body activity monitoring, mostly in the form of wearable off-the-shelf systems.

In the coming years, the sensors and systems that will be used in advanced industrial application will become smarter with built-in functions and embedded algorithms, such as machine learning and Kalman filters, which will be incorporated in the processing of data streams retrieved by IMUs, in order to increase their functionality and present a substitute for highly accurate (and expensive) camera-based MoCap systems. Furthermore, systems are expected to become smaller and portable in order to interfere less with the workers and workplace, while real-time (bio)feedback should accompany health and safety applications in order to aid in the adoption and acceptance of such technologies by industry workers. Marker-less MoCap systems, such as the Kinect, are low cost and offer adequate accuracy for certain classification and activity tracking tasks; however, attempts should be made for the further development and commercial distribution of processing algorithms that would improve their ease of use and capability to carry out data processing tasks. Optoelectronics have been widely and consistently used in robotics over the recent years, particularly in the research field of collaborative systems and are shown to increase the safety of human operators. In the future, the price drop of optoelectronic sensors and the release of more compact and easier to implement hybrid and data fusion solutions, as well as next-generation wearable lens-less cameras [83,84,85], will lead to fewer obstructions in jobsites and improve the practicality of camera-based approaches in other industry sectors.

## Figures and Tables

**Figure 1 sensors-20-05687-f001:**
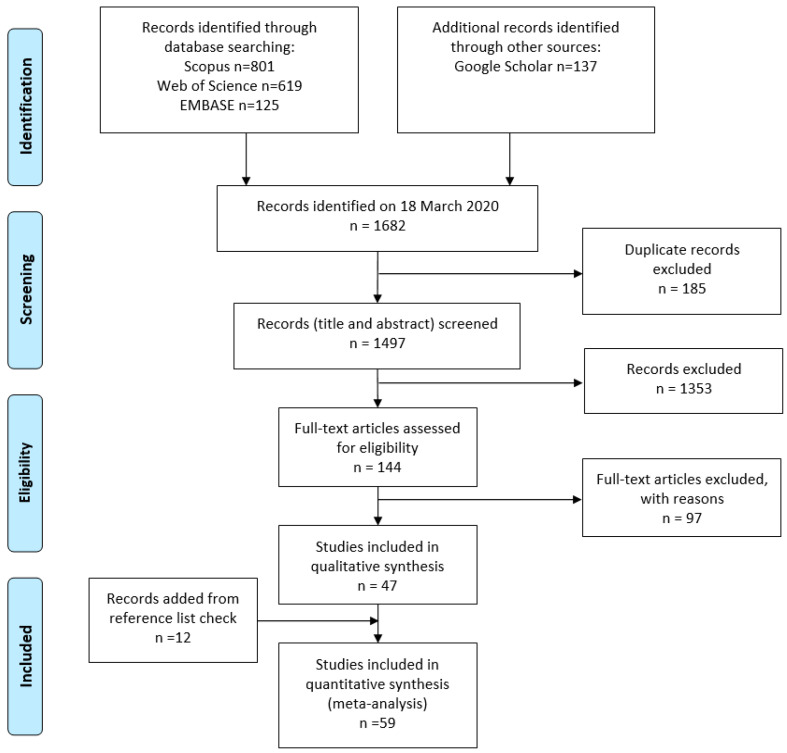
Search strategy Preferred Reported Item for Systematic review and Meta-Analysis (PRISMA) flow chart.

**Figure 2 sensors-20-05687-f002:**
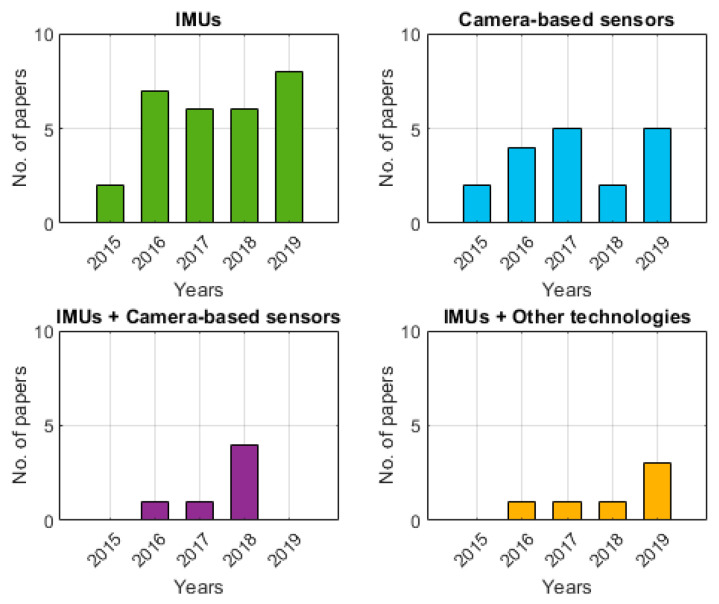
Number of publications per year divided by type of MoCap technology adopted.

**Figure 3 sensors-20-05687-f003:**
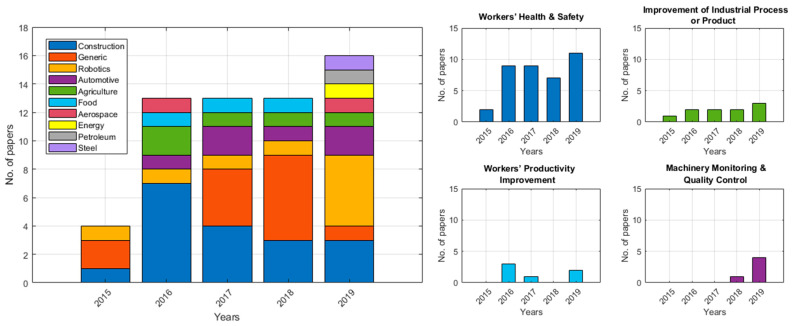
Number of publications per year and type of industry sector (**left**) and application (**right**).

**Figure 4 sensors-20-05687-f004:**
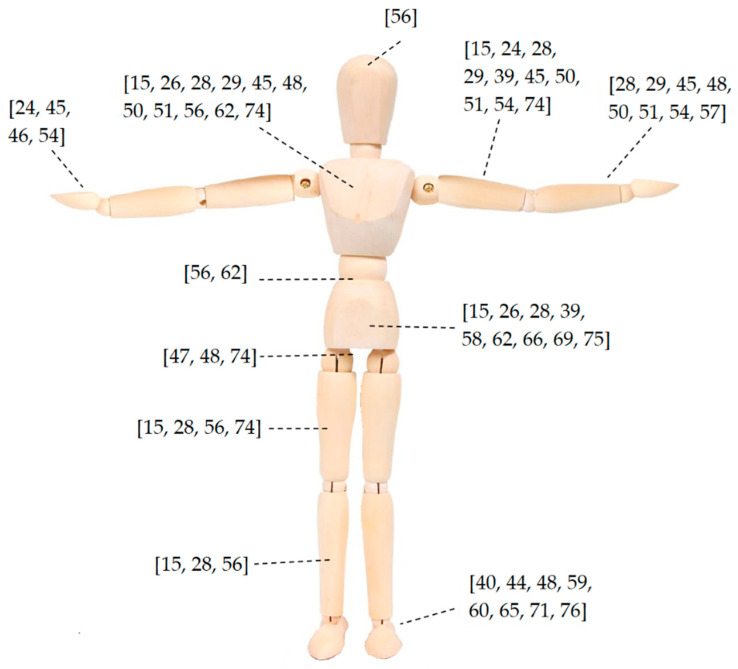
IMUs placement in the reviewed studies. Pre-developed models are excluded.

**Figure 5 sensors-20-05687-f005:**
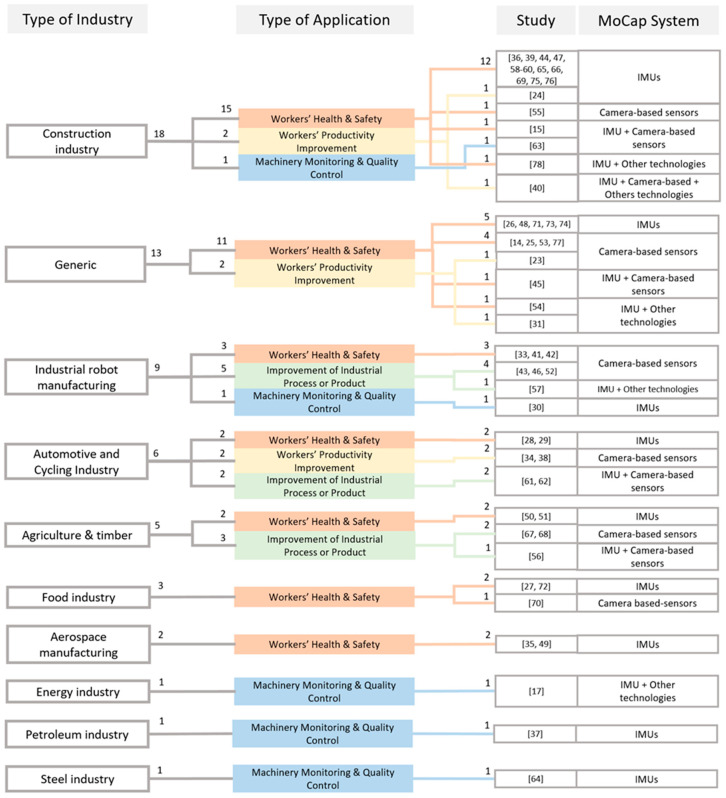
The relations between type of industry, application and MoCap system, for each considered study. Indexes at the top of each branch specify the number of studies associated to each block.

**Table 1 sensors-20-05687-t001:** Risk and quality assessment questions of the modified AXIS tool.

Question Number	AXIS Question Code	INTRODUCTION
Q1	1	Were the aims/objectives of the study clear?
		METHODS
Q2	2	Was the study design appropriate for the stated aim(s)?
Q3	3, 4 and 5	Was the sample size justified, clearly defined, and taken from an appropriate population?
Q4	8	Were the outcome variables measured appropriate to the aims of the study?
Q5	9	Were the outcome variables measured correctly using instruments/measurements that had been trialled, piloted or published previously?
Q6	10	Is it clear what was used to determined statistical significance and/or precision estimates? (e.g., *p*-values, confidence intervals)
Q7	11	Were the methods sufficiently described to enable them to be repeated?
		RESULTS
Q8	12	Were the basic data adequately described?
Q9	16	Were the results presented for all the analyses described in the methods?
		DISCUSSION
Q10	17	Were the authors’ discussions and conclusions justified by the results?
Q11	18	Were the limitations of the study discussed?
		OTHER
Q12	19	Were there any funding sources or conflicts of interest that may affect the authors’ interpretation of the results?

**Table 2 sensors-20-05687-t002:** Risk of bias assessment.

Risk of Bias	Score	Study	Number of Studies
High	0–6	-	0
Medium	7	[24,25]	2
Medium	8	[23,26,27,28,29,30,31,32,33]	8
Medium	9	[34,35,36,37,38,39,40,41,42,43,44,45]	12
Low	10	[14,15,36,46,47,48,49,50,51,52,53,54,55,56,57,58,59,60]	19
Low	11	[17,61,62,63,64,65,66,67,68,69,70]	11
Low	12	[71,72,73,74,75,76,77]	7

**Table 3 sensors-20-05687-t003:** Sensors used in the reported studies.

Sensors	Study	Number of Studies	Percentage of Studies
IMUs	[24,26,27,28,29,30,35,36,37,39,44,47,48,49,50,51,58,59,60,64,65,66,69,71,72,73,74,75,76]	29	49.2%
Camera-based Sensors *	[14,23,25,32,33,34,38,41,42,43,46,52,53,55,67,68,70,77]	18	30.5%
IMUs + Camera-based Sensors	[15,45,56,61,62,63]	6	10.2%
IMUs + Other Technologies	[17,31,54,57,78]	5	8.5%
IMUs + Camera-based + Other Technologies	[40]	1	1.7%

* Camera-based sensors include RGB, infrared, depth or optical cameras.

**Table 4 sensors-20-05687-t004:** Characteristics of the most used MoCap systems.

	Sensors		
	IMUs	Camera-Based	
		Marker-Based	Marker-Less
Accuracy	High (0.75° to 1.5°) ^3^	Very high (0.1 mm and 0.5°) ^1^; subject to number/location of cameras	Low (static, 0.0348 m [79]) subject to distance from camera
Set up	Straightforward; subject to number of IMUs	Requires time-consuming and frequent calibrations	Usually requires checkerboard calibrations
Capture volumes	Only subject to distance from station (if required)	Varies; up to 15 × 15 × 6 m ^1^	Field of view: 70.6° × 60°; 8 m depth range ^5^
Cost of installation	From USD 50 per unit to over USD 12,000 for a full-body suit ^4^	Varies; from USD 5000 ^2^ to USD 150,000 ^1^	USD 200 ^5^ per unit
Ease of use and data processing	Usually raw sensor data to ASCII files	Usually highly automated, outputs full 3D kinematics	Requires custom-made processing algorithms
Invasiveness (individual)	Minimal	High (markers’ attachment)	Minimal
Invasiveness (workplace)	Minimal	High (typically, 6 to 12 camera systems)	Medium (typically, 1 to 4 camera systems)
Line-of-sight necessity	No	Yes	Yes
Portability	Yes	Limited	Yes
Range	Usually up to 20 m from station ^3^ (if wireless)	Up to 30 m camera-to-marker ^1^	Low: skeleton tracking range of 0.5 m to 4.5 m ^5^
Sampling rate	Usually from 60 to 120 Hz ^3^ (if wireless)	Usually up to 250 Hz ^1^ (subject to resolution)	Varies; 15–30 Hz ^5^ or higher for high-speed cameras
Software	Usually requires bespoke or off-the-shelf software	Requires off-the-shelf software	Requires bespoke software, off-the-shelf solutions not available
Noise sources and environmental interference	Ferromagnetic disturbances, temperature changes	Bright light and vibrations	IR-interference with overlapping coverage, angle of observed surface
Other limitations	Drift, battery life, no direct position tracking	Camera obstructions	Camera obstructions, difficulties tracking bright or dark objects
Favoured applications	Activity recognition [31], identification of hazardous events/poses [44,58,60,65,66,69,75]	Human–robot collaboration [42], robot trajectory planning [52]	Activity tracking [34], gesture or pose classification [25,45,53]

^1^ Based on a sample layout with 24 Prime^x^41 Optritrack cameras. ^2^ Based on a sample layout with 4 Flex 3 Optritrack cameras. ^3^ Based on the specs of the Xsens MTW Awinda. ^4^ Based on the Xsens MVN. ^5^ Based on the Kinect V2.

**Table 5 sensors-20-05687-t005:** Types of industry sectors directly or indirectly suggested as potential recipients for the MoCap solutions developed in the reviewed works.

Industry	Study	Number of Studies	Percentage of Studies
Construction Industry	[15,24,36,39,40,44,47,55,58,59,60,63,65,66,69,75,76,78]	18	30.5%
Generic	[14,23,25,26,31,45,48,53,54,71,73,74,77]	13	22.0%
Industrial Robot Manufacturing	[30,32,33,41,42,43,46,52,57]	9	15.3%
Automotive and Cycling Industry	[28,29,34,38,61,62]	6	10.2%
Agriculture and Timber	[50,51,56,67,68]	5	8.5%
Food Industry	[27,70,72]	3	5.1%
Aerospace Manufacturing	[35,49]	2	3.4%
Energy Industry	[17]	1	1.7%
Petroleum Industry	[37]	1	1.7%
Steel Industry	[64]	1	1.7%

**Table 6 sensors-20-05687-t006:** Generic MoCap applications in industry.

Applications	Study	Number of Studies	Percentage of Studies
Workers’ Health and Safety	[14,15,25,26,27,28,29,33,35,36,39,41,42,44,45,47,48,49,50,51,53,54,55,58,59,60,65,66,69,70,71,72,73,74,75,76,77,78]	38	64.4%
Improvement of Industrial Process or Product	[32,43,46,52,56,57,61,62,67,68]	10	17.0%
Workers’ Productivity Improvement	[23,24,31,34,38,40]	6	10.1%
Machinery Monitoring and Quality Control	[17,30,37,63,64]	5	8.5%

**Table 7 sensors-20-05687-t007:** Machine learning classification approaches.

Study	Machine Learning Model	Input Data	Training Dataset	Classification Output	Accuracy
[24]	ANN, k-NN	Magnitude of linear and angular acceleration.	Manually labelled video sequences	Activity recognition	94.11%
[71]	SVM	Eight different motion components (2D position trajectories, profile magnitude of vel., acc. and jerk, angle and velocity).	2000 sample data points manually labelled	2 Fatigue states (Yes or No)	90%
[63]	Bayes classifier	Acceleration.	Labelled sensor data features	Type of landmarks (lift, staircase, etc.)	96.8% *
[64]	ANN	Cutting speed, feed rate, depth of cut, and the three peak spectrum amplitudes from vibration signals in 3 directions.	Labelled cutting and vibration data from 9 experiments	Worn tool condition (Yes or No)	94.9%
[36]	k-NN, MLP, RF, SVM	Quaternions, three-dimensional acceleration, linear velocity, and angular velocity.	Manually labelled video sequences	14 Activities (e.g., bending-up and bending-down)	Best RF, with 79.83%
[25]	RF	Joint angles evaluated from an artificial model built on a segmentation from depth images.	Manually labelled video sequences	5 different postures	87.1%
[47]	ANN	Three-dimensional acceleration.	Labelled dataset	Walk/slip/trip	94%
[53]	RF	Depth Comparison Features (DCF) from depth images.	Labelled dataset of 5000 images	7 RULA score postures	93%
[57]	DAG-SVM	Rotation angle and EMG signals.	Dataset acquired with known object weight	Light objects, large objects, and heavy objects	96.67%
[69]	One-class SVM	Acceleration.	Dataset of normal walk samples	2 states (walk, near-miss falls)	86.4%
[32]	CNN	Distance of nearest point, curvature and HSV colour.	Pre-existing dataset of objects from YOLO [80]	Type of object	97.3% *

* Combined accuracy of the classification process and the success rate of the task. Abbreviations used: ANN = Artificial Neural Network; k-NN = k-Nearest Neighbour; SVM = Support Vector Machine; MLP = Multilayer Perceptron; DAG = Directed Acyclic Graph; CNN = Convolutional Neural Network; HSV = Hue Saturation Value.

**Table 8 sensors-20-05687-t008:** Summary of study designs and findings.

Study	Sample Size	Sensor Placement	Number of IMUs	MoCap Systems and Complementary Sensors	Study Findings
[34]	1	-		6 × Kinect, **16 × ARTtrack2 IRCs**	Evaluation of workers’ walk paths in assembly lines showed an average difference of 0.86 m in the distance between planned and recorded walk paths.
[24]	1	Upper Body	1	-	Activity recognition of over 90% between different construction activities; very good accuracy in activity duration measurements.
[61]	8	-	3	16 × VICON IRCs (50 × passive markers), 1 × pressure sensor, 2 × belt load cells	Both tested car crash test surrogates had comparable overall ISO scores; THOR-M scored better in acceleration and angular velocity data; Hybrid III had higher average ISO/TS 18571 scores in excursion data.
[26]	11	Upper Body	2	Heart rate, skin temperature, air humidity, temperature and VO2	IMUs can discriminate rest from work but they are less accurate differentiating moderate from hard work. Activity is a reliable predictor of cold stress for workers in cold weather environments.
[71]	20	Lower Body	1	-	Fatigue detection of workers with an accuracy of 80%.
[72]	15	Upper body	1	1 × electronic algometer	Vineyard-workers spent more than 50% of their time with the trunk flexed over 30°. No relationship between duration of forward bending or trunk rotation and pain intensity.
[23]	1	-	-	4 × Kinect	Proof of concept of a MoCap system for the evaluation of the human labour in assembly workstations.
[27]	5	Full body	17	-	Workers lifting kegs from different heights showed different torso flexion/extension angles.
[28]	2	Full body	7	-	Proof of concept of an IMU system for workers’ postural analyses, with an exemplary application in automotive industry.
[62]	20	Upper body	3	4 × Optotrack IRCs (active markers), 2 × Xsensor pressure pads	No significant differences in terms of body posture between the tested truck seats; peak and average seat pressure was higher with industry standard seats; average trunk flexion was higher with industry standard seats by 16% of the total RoM.
[78]	3	Upper body	17	1 × UWB, 1 × Perception Neuron, 1 × phone camera	In construction tasks, the accuracy of the automatic safety risk evaluation was 83%, as compared to the results of the video evaluation by a safety expert.
[17]	-	Machinery	1	1 × mechanical scanning sonar	The position tracking accuracy of remotely operated vehicles in a nuclear power plant was within centimetre level when compared to a visual positioning method.
[73]	16	Full body	17	**1 × Certus Optotrak**, 6 × Kistler FPs, 2 × Xsens instrumented shoes	The root-mean square differences between the estimated and measured hand forces during manual materials handling tasks from IMUs and instrumented force shoes ranged between 17-21N.
[81]	1	Full body	-	4 × depth cameras	Proof of concept of a motion analysis system for the evaluation of the human labour in assembly workstations.
[35]	10	Full body	17	-	Demonstrated an IMU MoCap system for the evaluation of workers’ posture in the aerospace industry.
[29]	1	Upper body	4	1 × wearable camera, 3 × video cameras 6 × BTS EMG sensors	Proof of concept of an EMG and IMU system for the risk assessing of workers’ biomechanical overload in assembly lines.
[74]	625	Full body	4	-	More time spent in leisure physical activities was associated with lower pain levels in a period of over 12 months. Depending on sex and working domain, high physical activity had a negative effect on the course of pain over 12 months.
[63]	-	Machinery	1	1 × mobile phone camera	Three-dimensional localization of distant target objects in industry with an average position errors of 3% in the location of the targeted objects.
[64]	-	Machinery	1	-	Tool wear detection in CNC machines using an accelerometer and an artificial neural network with an accuracy of 88.1%.
[65]	8	Lower Body	1	1 × video camera	Distinguish low-fall-risk tasks (comfortable walking) from high-risk tasks (carrying a side load or high-speed walking) in construction workers walking on I-beams.
[75]	10	Upper body	1	1 × force plate	Wearing a harness loaded with common iron workers’ tools could be considered as a moderate fall-risk task, while holding a toolbox or squatting as a high-risk task.
[46]	1	Upper body	-	1 × Kinect	Teleoperation of a robot’s end effector through imitation of the operator’s arm motion with a similarity of 96% between demonstrated and imitated trajectories.
[66]	10	Upper body	1	-	The Shapiro–Wilk statistic of the used acceleration metric can distinguish workers’ movements in hazardous (slippery floor) from non-hazardous areas.
[76]	16	Lower Body	1	-	The gait stability while walking on coated steel beam surfaces is greatly affected by the slipperiness of the surfaces (*p* = 0.024).
[36]	1	Full body	17	1 × video camera	Two IMU sensors on hip and either neck or head showed similar motion recognition accuracy (higher than 0.75) to a full body model of 17 IMUs (0.8) for motion classification.
[25]	8	Full body	-	1 × Kinect	Posture classification in assembly operations from a stream of depth images with an accuracy of 87%; similar but systematically overestimated EAWS scores.
[47]	3	Lower Body	1	-	Identification of slip and trip events in workers’ walking using an ANN and phone accelerometer with detection accuracy of 88% for slipping, and 94% for tripping.
[30]	-	Machinery	2	-	High frequency vibrations can be detected by common accelerometers and can be used for micro series spot welder monitoring.
[31]	1	Full Body	17	E-glove from Emphasis Telematics	Measurements from the IMU and force sensors were used for an operator activity recognition model for pick-and-place tasks (precision 96.12%).
[48]	8	Full Body	4	1 × ECG	IMUs were a better predictor of fatigue than ECG. Hip movements can predict the level of physical fatigue.
[37]	-	Machinery	1	-	The orientation of a robot (clock face and orientation angles) for pipe inspection can be estimated via an inverse model using an on-board IMU.
[77]	12	Full Body	-	**Motion analysis system (45 × passive markers)**, 2 × camcorders	The 3D pose reconstruction can be achieved by integrating morphological constraints and discriminative computer vision. The performance was activity-dependent and was affected by self and object occlusion.
[49]	3	Full Body	17	1 × manual grip dynamometer, 1 × EMG	Workers in a banana harvesting adapt to the position of the bunches and the used tools leading to musculoskeletal risk and fatigue.
[50]	2	Upper Body	8	6 × EMG	A case study on the usefulness of the integration of kinematic and EMG technologies for assessing the biomechanical overload in production lines.
[51]	2	Upper Body	8	6 × EMG	Demonstration of an integrated EMG-IMU protocol for the posture evaluation during work activities, tested in an automotive environment.
[52]	-	Machinery	-	4 × IRCs (5 × passive markers)	Welding robot path planning with an error in the trajectory of the end-effector of less than 3 mm.
[38]	42	-	-	1 × Kinect	The transfer of assembly knowledge between workers is faster with printed instructions rather with the developed smart assembly workplace system (*p*-value = 7 × 10^−9^) as tested in the assembly of a bicycle e-hub.
[53]	-	-	-	1 × Kinect	Real time RULA for the ergonomic analysis for assembly operations in industrial environments with an accuracy of 93%.
[39]	-	-	3	1 × Kinect, 1 × Oculus rift	Smartphone sensors to monitor workers’ bodily postures, with errors in the measurements of trunk and shoulder flexions of up to 17˚.
[67]	-	-	-	1 × Kinect	Proof of concept of a real-time MoCap platform, enabling workers to remotely work on a common engineering problem during a collaboration session, aiding in collaborative designs, inspection and verifications tasks.
[40]	-	-	1	1 × CCTV, radio transmitter	A positioning system for tracking people in construction sites with an accuracy of approximately 0.8 m in the trajectory of the target.
[54]	10	Upper Body	3	EMG	A wireless wearable system for the assessment of work-related musculoskeletal disorder risks with a 95% and 45% calculation accuracy of the RULA and SI metrics, respectively.
[14]	12	-	-	1 × Kinect, **15 × Vicon IRC (47 × passive markers)**	RULA ergonomic assessment in real work conditions using Kinect with similar computed scores compared to expert observations (*p* = 0.74).
[68]	-	-	-	2 × Kinect, 1 × laser motion tracker	Digitising the wheel loading process in the automotive industry, for tracking the moving wheel hub with an error less than the required assembly tolerance of 4 mm.
[41]	-	-	-	1 × Kinect, 1 × Xtion	A demonstration of a real time trajectory generation algorithm for human–robot collaboration that predicts the space that the human worker can occupy within the robot’s stopping time and modifies the trajectory to ensure the worker’s safety.
[42]	-	Upper Body	-	1 × OptiTrack V120: Trio system	A demonstration of a collision avoidance algorithm for robotics aiming to avoid collisions with obstacles without losing the planned tasks.
[55]	-	Lower Body	-	1 × Kinect, 1 × bumblebee XB3 camera, 1 × 3D Camcoder, **2 × Optotrack IRCs**, 1 × Goniometer	A vision-based and angular measurement sensor-based approach for measuring workers’ motions. Vision-based approaches had about 5–10 degrees of error in body angles (Kinect’s performance), while an angular measurement sensor-based approach measured body angles with about 3 degrees of error during diverse tasks.
[43]	-	-	-	1 × Oculus Rift, 2 × PlayStation Eye cameras	Using the Oculus rift to control remote robots for human computer interface. The method outperforms the mouse in rise time, percent overshoot and settling time.
[56]	29	Machinery and Upper Body	6	11 × IRC Eagle Digital	Proof of concept method for the evaluation of sitting posture comfort in a vehicle.
[45]	-	Upper Body	7	1 × Kinect	A demonstration of a human posture monitoring systems aiming to estimate the range of motion of the body angles in industrial environments.
[33]	-	Upper Body	-	1 × HTC Vive system	Proof of concept of a real-time motion tracking system for assembly processes aiming to identify if the human worker body parts enter the restricted working space of the robot.
[15]	6	Full Body	8	1 × video Camera	A demonstration of a system for the identification of detrimental postures in construction jobsites.
[57]	10	Upper Body	1	8 × EMG	A wearable system for human–robot collaborative assembly tasks using hand-over intentions and gestures. Gestures and intentions by different individuals were recognised with a success rate of 73.33% to 90%.
[58]	2	Upper Body	1	-	Detection of near-miss falls of construction workers with 74.9% precision and 89.6% recall.
[69]	5	Upper Body	1	-	Automatically detect and document near-miss falls from kinematic data with 75.8% recall and 86.4% detection accuracy.
[44]	4	Lower Body	2	video cameras	A demonstration of a method for detecting jobsite safety hazards of ironworkers by analysing gait anomalies.
[60]	9	Lower Body	1	video cameras	Identification of physical fall hazards in construction, results showed a strong correlation between the location of hazards and the workers’ responses (0.83).
[59]	4	Lower Body	2	Osprey IRC system	Distinguish hazardous from normal conditions on construction jobsites with 1.2 to 6.5 mean absolute percentage error in non-hazard and 5.4 to 12.7 in hazardous environments.
[32]	-	-	-	1 × video camera	Presentation of a robot vision system based on CNN and a Monte Carlo algorithm with a success rate of 97.3% for the pick-and-place task.
[70]	15	-	-	1 × depth camera	A system aiming to warn a person while washing hands if improper application of soap was detected based on hand gestures, with 94% gesture detection accuracy.

**In Bold: Validation Systems.** IRC: infrared camera; THOR-M: test device for human occupant restraint; VO2: oxygen consumption; FP: force plate; EAWS: European assembly worksheet; ECG: electrocardiogram; CNC = Computer Numerical Control.

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
