# Peer review of "Motion Capture Technology in Industrial Applications: A Systematic Review"

_sensors, 2020, doi:10.3390/s20195687_

Round 1

Reviewer 1 Report

The paper is well written, clear and well supported. Interest is strong and should be of interest to the scientific community. The quality of the articles appraised with the AXIS tool is very timely.

However, the article suffers from several points:

-Even if the study of the last 5 years is very well conducted and very rigorous, we don't see the recommendations that the authors draw from this research. Which way forward? A table of advantages and disadvantages of the different techniques presented should be proposed?

-a comparison should be made of the performance of MoCap and IMU systems in terms of measurement accuracy, errors, calibration, bias introduced by placement or positioning, etc.

-What should we learn from this study if researchers wanted to make a contribution? What are the ways and prospects for research in the field?

-Wearable systems are now widely used in the health sector for remote monitoring of physiological parameters, activities, etc. Why is this area not clearly visible in the application field? Give some concrete applications?

- Why the review papers and conference abstracts were all excluded from this work? What is the reason? It's not argued.

Table 7 "Summery of study designs and findings" replace to "Summary..."

Table 7 and Figure 5 are the result of an excellent work. Congratulations. But this work needs to be argued and synthesized. What should we learn?

The authors said that "the hybrid systems were found to have improved tracking performance in noisy and highly dynamic industrial environments..." or "fewer IMUs placed on specific anatomical sections (e.g. hip and neck) showed better task classification performance than a greater number of IMUs distributed on the entire body". It would be useful to give some values of these improvements and classification performance.

Generally speaking, you remained too qualitative and not quantitative enough.

In conclusion, the approach proposed by the authors on the basis of this literature review could be presented in a specific section. What is the best approach? What are the recommendations?

Finally, the authors acknowledge the limitations of this review which stem from the potential publication bias and selective reporting across studies and from the short review period that narrowed the analysis of trends in a period of only five years. Why this choice of 5 years?

Author Response

Reviewer 1

The authors would like to thank the reviewers for their thorough review and encouraging comments, as we believe that their input has greatly improved our manuscript. We hope that the amendments listed here have positively addressed all their comments. Please also note that all changes in the manuscript are in bold.

The paper is well written, clear and well supported. Interest is strong and should be of interest to the scientific community. The quality of the articles appraised with the AXIS tool is very timely. However, the article suffers from several points:

  • Even if the study of the last 5 years is very well conducted and very rigorous, we don't see the recommendations that the authors draw from this research. Which way forward? A table of advantages and disadvantages of the different techniques presented should be proposed?

Thank you for your kind words and for this comment. We added in our results (lines 190-198) a table of advantages/disadvantages of IMU, marker-based and marker-less sensors that reports on multiple aspects of each type of sensor (costs, accuracy, impact in the lab layout, training, ease of processing, etc):

Considering that the most frequently adopted sensors used in industry were IMUs (e.g. Xsens MVN) and marker-based or marker-less (e.g. Kinect) camera systems, their characteristics, advantages and disadvantages were also captured (Table 4) in order to evaluate how each sensors type is appropriate to the different applications. Naturally, the characteristics of each system vary greatly depending on the number, placement, settings and calibration requirements of the sensors, yet, general recommendations can be made for the adoption of a particular type of sensor for distinct tasks. Additionally, given the required level of accuracy, capture volume, budget and workplace limitations or other considerations, Table 4 shows the specifications and most favoured industrial applications for each type of sensor (e.g. activity recognition, or human-robot collaboration).”

Table 4 Advantages and disadvantages of the most used MoCap systems

Sensors

IMUs

Camera-based

Marker-based

Marker-less

Accuracy

High (0.75° to 1.5°)3

Very high (0.1mm & 0.5°)1; subject to number/location of cameras

Low (static, 0.0348m [79]) subject to distance from camera

Set up

Straightforward; subject to number of IMUs

Requires time-consuming and frequent calibrations

Usually requires checkerboard calibrations

Capture volumes

Only subject to distance from station (if required)

Varies; up to 15 × 15 × 6m1

Field of view: 70.6° x 60°; 8m depth range5

Cost of installation

Varies; from 50$ per unit to over 12,000$ for a full-body suit4

Varies; from 5,000 $2 to 150,000 $1

200$5 per unit

Ease of use and data processing

Usually raw sensor data to ASCII files

Usually highly automated, outputs full 3D kinematics

Requires custom-made processing algorithms

Invasiveness (individual)

Minimal

High (markers’ attachment)

Minimal

Invasiveness (workplace)

Minimal

High (typically, 6 to 12 camera systems)

Medium (typically, 1 to 4 camera systems)

Line-of-sight necessity

No

Yes

Yes

Portability

Yes

Limited

Yes

Range

Usually up to 20m from station3 (if wireless)

Up to 30m camera-to-marker1

Low: skeleton tracking range of 0.5m to 4.5m5

Sampling rate

Usually from 60 to 120 Hz3 (if wireless)

Usually up to 250 Hz1 (subject to resolution)

Varies; 15-30Hz5 or higher for high speed cameras

Software

Usually requires bespoke or off-the-shelf software

Requires off-the-shelf software

Requires bespoke software, off-the-shelf solutions not available

Noise sources and environmental interference

Ferromagnetic disturbances, temperature changes

Bright light and vibrations

IR-interference with overlapping coverage, angle of observed surface

Other limitations

Drift, battery life, no direct position tracking

Camera obstructions

Camera obstructions, difficulties tracking bright or dark objects

Favored applications

Activity recognition [31], identification of hazardous events and poses [44, 58, 60, 65, 66, 69, 75]

Human-robot collaboration [42], robot trajectory planning [52]

Activity tracking [34], gesture or pose classification [25, 45, 53]

1Based on a sample layout with 24 Primex41 Optritrack cameras.

2Based on a sample layout with 4 Flex 3 Optritrack cameras.

3Based on the specs of the Xsens MTW Awinda.

4Based on the Xsens MVN

5Based on the Kinect V2

Also, based on the above-mentioned information, further recommendations were added in the discussion as regards the future trends in MoCap solution for the industry (lines 400-416):

In the coming years, the sensors and systems that will be used in advanced industrial application will become smarter with built-in functions and embedded algorithms, such as machine learning and Kalman filters, which will be incorporated in the processing of data streams retrieved by IMUs, in order to increase their functionality and present a substitute for highly accurate (and expensive) camera-based MoCap systems. Furthermore, systems are expected to become smaller and portable in order to interfere less with the workers and workplace, while real-time (bio)feedback should accompany health & safety applications in order to aid in the adoption and acceptance of such technologies by industry workers. Marker-less MoCap systems, such as the Kinect, are low-cost and offer adequate accuracy for certain classification and activity tracking tasks, however, attempts should be made for the further development and commercial distribution of processing algorithms that would improve their ease of use and capability to carry out data processing tasks. Optoelectronics have been widely and consistently used in robotics over the recent years, particularly in the research field of collaborative systems and are shown to increase the safety of human operators. In the future, the price drop of optoelectronic sensors and the release of more compact and easier to implement hybrid and data fusion solutions, as well as next-generation wearable lens-less cameras [83-85], will lead to less obstructions in jobsites and improve the practicality of camera-based approaches in other industry sectors.”

  • A comparison should be made of the performance of MoCap and IMU systems in terms of measurement accuracy, errors, calibration, bias introduced by placement or positioning, etc.

Kindly refer to the 1st comment and Table 4 for a comparison of MoCap systems in terms of accuracy, errors, and other sources of bias. Also, we added another relevant citation in text that may help the reader compare commercially available solution in terms of accuracy (lines 368-371): “As regards the use of commercially available MoCap solutions, a comparison was made of their limitations, advantages and applicability to industrial applications (Table 4) while the accuracy of off-the-shelf MoCap systems has been also extensively reviewed by van der Kruk and Reijne [83].

  • What should we learn from this study if researchers wanted to make a contribution? What are the ways and prospects for research in the field?

Thank you for highlighting that this important aspect of the review wasn’t clear/discussed enough. We partially addressed these questions in comment 1, by introducing and discussing a new table of advantages and disadvantage (Table 7), and in lines 396-411. Further suggestions for future works are now also discussed in lines 353-361: These findings highlighted how the research activity on wearable systems for industrial applications is going towards solutions that can be easily embedded in working cloths. Improving key factors such as wearability, washability, battery duration, data storage and edge computing will be therefore essential. This improvement in the hardware design will have a direct impact on the amount and the quality of the data collection. This, as well, will have a beneficial effect on software development, especially for machine learning applications, were huge quantity of data are required. In this regards, attempts should be made for the further development and commercial distribution of processing algorithms that would improve the ease of use of such systems and the data processing.”

  • Wearable systems are now widely used in the health sector for remote monitoring of physiological parameters, activities, etc. Why is this area not clearly visible in the application field? Give some concrete applications?

A very keen comment, thank you for pointing this out. This topic is now addressed in text (lines 319-328): “Workers’ health & safety was found to be the most prolific research area. Even though wearable sensors are widely used in clinical settings for the remote monitoring of physiological parameters (e.g. heart rate, blood pressure, body temperature, VO2), only a single study [26] has employed multiple sensors for the measurement of such metrics in industrial scenarios. This can be attributed to the industries involved being interested in the prevention of work-related incidents that can lead to absence from work, rather than in the normative function of the workers’ body. As anticipated, health & safety research focused on the most common musculoskeletal conditions (e.g. back pain) and injuries (e.g. trips or injuries due to bad body posture), while the industries in which workers deal with heavy biomechanical loads or high risk of accidents (e.g. construction, Table 5) were the industries that drove the research. Fatigue and postural distress were also successfully detected by wearable inertial MoCap technologies…”

  • Why the review papers and conference abstracts were all excluded from this work? What is the reason? It's not argued.

Prior to data extraction, predefined tables were created for the collection of info such as participants’ numbers, and sensors’ type, numbers and placement (lines 131-132): “Data from all considered articles were extracted … using predefined tables.” Therefore, review papers and conferences abstracts were excluded since, inherently, they do not contain all the necessary info for the construction of such tables; this is argued later in text (lines 103-105): “Articles were included only if the participants’ sample size (where applicable), and the type, number and placement of all used sensors were reported.” With reference to your comment, an additional justification was also included along with the study’s inclusion and exclusion criteria (lines 94-95): “Review papers and conference abstracts were all excluded from this work since they do not typically report on all elements of the predefined tables that were used for data extraction for the purposes of this review.”

Equally importantly, it was not possible to access with the AXIS protocol the quality of a work based only on an abstract, therefore inclusion of conference abstracts was not appropriate.

  • Table 7 "Summery of study designs and findings" replace to "Summary..."

“Summery” changed to “Summary”, thank you.

  • Table 7 and Figure 5 are the result of an excellent work. Congratulations. But this work needs to be argued and synthesized. What should we learn?

Table 7 (now Table 8) and Figure 5 are now better described in the discussion (lines 294-303): “Findings suggest that the selection of the optimal MoCap system to adopt was primarily driven by the type of application (Figure 5); for instance, monitoring and quality control was mainly achieved via IMUs sensors, while productivity improvement via camera-based (marker-less) systems. Type of industry was the second factor that had an impact on the choice of a MoCap system (Figure 5); for example, in highly dynamic environment, like in construction sites, where the setup and configuration of the physical space change over time, wearable inertial sensors were the best option, since they could ensure a robust and continuous assessment of the operator’s health and safety. In industrial robot manufacturing instead, where the environmental constraints are known and constant, health and safety issues were primarily addressed by camera-based systems.”

  • The authors said that "the hybrid systems were found to have improved tracking performance in noisy and highly dynamic industrial environments..." or "fewer IMUs placed on specific anatomical sections (e.g. hip and neck) showed better task classification performance than a greater number of IMUs distributed on the entire body". It would be useful to give some values of these improvements and classification performance.

Thank you for pointing this out. To reinforce these statements, numerical values were added in the text (lines 313-314). “For instance, in Papaioannou, et al. [40], the trajectory tracking error caused by occlusion in a hybrid system was approximately half that of a camera-based tracking system.” And lines 332-334: “In Kim et al. [36] a task classifier based on just two IMUs on the hip and head of the subject reached an accuracy of 0.7617 against the 0.7983 of the classifier based on 17 IMUs placed on the entire body.”

  • Generally speaking, you remained too qualitative and not quantitative enough. In conclusion, the approach proposed by the authors on the basis of this literature review could be presented in a specific section. What is the best approach? What are the recommendations?

Kindly refer to comments 7 and 3, and lines 400-415.

  • Finally, the authors acknowledge the limitations of this review which stem from the potential publication bias and selective reporting across studies and from the short review period that narrowed the analysis of trends in a period of only five years. Why this choice of 5 years?

We opted for a time frame that could return enough papers in order to justify conclusions but not an excessive number that would hinder the analysis and reporting of findings. In the mature field of motion capture analysis, a search on a period of 5 years returned more than 1600 records (figure 1) which was deemed adequate during our preliminary searches. In fact, after manuscript screening, almost 60 titles were summarised in table 7 and Figure 5, returning enough material to synthesize evidence and see trends (as depicted in figure 3), while also giving us the opportunity to report on multiple aspects of each work, such as all the used sensors and study findings (Table 7).

The choice of the time frame of 5 years is also supported now in text (lines 380-385): “Finally, another limitation at a review-level arises from the short review period that narrowed the reporting of findings in a period of five years; however, the selected review period returned an adequate number of records for the justification of conclusions and exposure of trends (e.g. Figure 3), while facilitating the reporting of multiple aspects of the reviewed articles, such as the studies’ design and key findings (Table 8).”

Reviewer 2 Report

The authors present a systematic review which explores the use of motion capture (MoCap) in industrial application. The review is of interest; however I do have some comments.

  1. How is this review different to previously published reviews. Are the authors present it from a different angle, or are they performing an ‘updated’ search?
  2. The authors cite ‘industrial application’, however more clarity is needed. It appears that any article that uses MoCap and is focused towards a topic is included. Is this correct?
  3. Was the systematic review pre-registered such as on the PROSPERO system (https://www.crd.york.ac.uk/prospero/)? If not, what did the authors do to ensure integrity in the search?
  4. Where were reviewed conference papers included?
  5. Further, why were white collar worker studies excluded? And what do the authors define as white collar?
  6. Were any inter rate reliability performed between the reviewers when performing title, abstract and full text screening?
  7. The authors state that theoretical proof of concept papers were excluded. However, how is this defined, studies such as [32], [22], [21] appear to be proof of concept (considering sample size of 1)
  8. The authors provide not limitations to the review itself, all systematic reviews have some limitations. Please consider highlight them.
  9. Overall the article is well written, but I find the discussion lacking in terms of synthesis, next steps for MoCap and gaps in the literature. Please consider adding these in.

Author Response

Reviewer 2

The authors would like to thank the reviewers for their thorough review and encouraging comments, as we believe that their input has greatly improved our manuscript. We hope that the amendments listed here have positively addressed all their comments. Please also note that all changes in the manuscript are in bold.

The authors present a systematic review which explores the use of motion capture (MoCap) in industrial application. The review is of interest; however, I do have some comments.

  • How is this review different to previously published reviews? Are the authors present it from a different angle, or are they performing an ‘updated’ search?

Thank you for this comment. To the authors knowledge, this is the first review to track the use of MoCap techniques in literature in a systematic way. This also reflects now in text (lines 66-68): “Previous reviews have focused on MoCap in robotics [13], clinical therapy and rehabilitation [18], computer animation [12], and sports [19]; however, the use of MoCap for industrial applications has not been yet recorded in a systematic way.”

  • The authors cite ‘industrial application’, however more clarity is needed. It appears that any article that uses MoCap and is focused towards a topic is included. Is this correct?

Your assumption is correct, we further clarified this in text (lines 96-100): “..only studies that actively employed sensors that are designed to directly measure motion (i.e. the position, displacement, velocity or acceleration of an object) for either primary and secondary industrial applications were included; in this context, an industrial application was defined as any process related to the extraction of raw materials (e.g. metals or farming), or the manufacturing and assembly of goods (e.g. cars or buildings)”.

  • Was the systematic review pre-registered such as on the PROSPERO system (https://www.crd.york.ac.uk/prospero/)? If not, what did the authors do to ensure integrity in the search?

Thank you for this insightful comment. Indeed, pre-registering systematic reviews is the recommended practice that helps avoid duplication of existing works and reduces the reporting bias. However, as per the Prospero and Prisma recommendations, our review may not be pre-registered (and in fact, it is not accepted for pre-registration); as per the Prospero requirements (available at: https://www.crd.york.ac.uk/prospero/#aboutpage), PROSPERO does not accept:

  1. Systematic reviews without an outcome of clear relevance to the health of humans.
  2. Literature reviews that use a systematic search

Therefore, since our review is a literature review that is systematic due to the methods that were used to collect data and given that our outcomes do not assess a clinical, diagnostic, epidemiological or other human health intervention, the authors didn’t deem appropriate or necessary to pre-register the study’s protocol.

  • Where were reviewed conference papers included?

All conference papers that met the inclusion criteria of the study were included in the paper (lines 91-92): “The search included both in press and issued articles that were published in scientific journals or conference proceedings alike.” For example, citations: [24], [31], [33], [37], [38], [42] are all conference papers. However, conference publications without a full text were excluded, as described now in text (lines 94-95): “conference abstracts were all excluded from this work since they do not typically report on all elements of the predefined tables that were used for data extraction for the purposes of this review.” Additionally, it is not possible to access with the AXIS protocol the quality of a work based only on an abstract, therefore inclusion of conference abstracts was not appropriate.

  • Further, why were white collar worker studies excluded? And what do the authors define as white collar?

The definition of white colour workers for the purposes of the study was included in the manuscript (lines 101-102): “… white collar workers (e.g. office or other non-manual workers)”.

To ensure consistency of the extracted information for comparison purposes, only studies that looked at either primary and secondary industrial applications were included, and therefore, the inclusion of works related to manufacturing processes and/or blue-colour workers (i.e. manual labour workers). After all, studies looking at white colour workers (people who typically perform administrative or managerial work), routinely assess the posture of such subjects working for prolonged hours while sited on a desk, while their findings are naturally irrespective of the type of industry that white-colour workers are employed.

  • Were any inter rate reliability performed between the reviewers when performing title, abstract and full text screening?

During title and abstract screening, both reviewers followed the same protocol for the inclusion and exclusion of the reviewed studies, and only the studies that were undoubtably non-compliant with the predefined criteria were excluded (e.g. clinical or animal studies), while the remaining works were all included for further screening. Then, during full text screening, all manuscripts were screened by both reviewers separately, thus ensuring excellent inter rate reliability: manuscripts that both reviewers agreed to include in the review were automatically considered for further analysis, while in the occasion of conflicting decisions, the reviewers debated until both parties agreed to a conclusion. For more detailed information, please also see lines 109-116.

  • The authors state that theoretical proof of concept papers were excluded. However, how is this defined, studies such as [32], [22], [21] appear to be proof of concept (considering sample size of 1)

Thank you for pointing this out. As correctly mentioned, we included all proof of concept studies that tested/validated their application with at least 1 subject. On the contrary, proof of concept papers that simply presented an untested idea or hardware without presenting any experimental testing (and therefore without actually using MoCap technologies for industrial purposes) were all excluded. Along these lines, we agree with the reviewer that the used term “theoretical proof of concept papers” was misleading and it is now changed to “proof of concept papers that were not tested experimentally”.

  • The authors provide not limitations to the review itself, all systematic reviews have some limitations. Please consider highlight them.

Limitation are presented in text (lines 372-385) and also here for convenience:

“Even though all the reviewed works were assessed as being prone to medium and low risks of bias individually (Table 2), the main limitation at a study level was that more than half of the reviewed works (51%) did not properly report funding and conflict sources. This, may be an indication of a critical source of bias, particularly in studies directly driven by the beneficiary industry, or in works that demonstrate MoCap systems that may be commercially available in the future. A limitation of this review stems from the potential publication bias and selective reporting across studies, which may affect the accumulation of thorough evidence in the field. Efforts from industry bodies to incorporate MoCap applications in their facilities that were either unsuccessful or were not disseminated in scientific journals were likely overlooked in this review. Finally, another limitation at a review-level arises from the short review period that narrowed the analysis of trends in a period of five years; however, the opted review period returned an adequate number of records for the justification of conclusions and exposure of trends (e.g. Figure 3), while facilitating the reporting of multiple aspects of the reviewed articles, such as the studies’ design and key findings (Table 8).”

  • Overall, the article is well written, but I find the discussion lacking in terms of synthesis, next steps for MoCap and gaps in the literature. Please consider adding these in.

Thank you for this comment. Kindly refer to the 1st comment regarding gaps in the literature, while further recommendations were added in the discussion as regards the future trends in MoCap solution for the industry (lines 396-411): In the coming years, the sensors and systems that will be used in advanced industrial application will become smarter with built-in functions and embedded algorithms, such as machine learning and Kalman filters, which will be incorporated in the processing of data streams retrieved by IMUs, in order to increase their functionality and substitute highly accurate camera-based MoCap systems. Furthermore, systems are expected to become smaller and portable in order to interfere less with the workers and workplace, while real-time (bio)feedback should accompany health & safety applications in order to aid in the adoption and acceptance of such technologies by industry workers. Marker-less MoCap systems, like the Kinect, are low-cost and offer adequate accuracy for certain classification and activity tracking tasks, however, attempts should be made for the further development and commercial distribution of processing algorithms that would improve their ease of use and data processing. Optoelectronics were widely and consistently used in robotics over the recent years, especially in the research field of collaborative systems and to increase the safety of human operators. In the future, the price drop of optoelectronic sensors and the release of more compact and easier to implement hybrid and data fusion solutions, as well as next-generation wearable lens-less cameras [83-85], will lead to less obstructions in jobsites and improve the practicality of camera-based approaches in other industry sectors.”

Round 2

Reviewer 1 Report

Thank you for your additions to respond to the many requests for more information.

Reviewer 2 Report

The authors have addressed my concerns.